# Artificial intelligence with temporal features outperforms machine learning in predicting diabetes

**Iqra Naveed[1], Muhammad Farhat Kaleem[1], Karim Keshavjee[2]\*, Aziz Guergachi[2,3,4]**

**1** Department of Electrical Engineering, University of Management and Technology, Lahore, Pakistan,
**2** Institute of Health Policy, Management and Evaluation, University of Toronto, Toronto, Canada, **3** Ted Rogers School of Information Technology Management, Toronto Metropolitan University, Toronto, Canada,
**4** Department of Mathematics and Statistics, York University, Toronto, Canada

\* karim.keshavjee@utoronto.ca

**Data Availability Statement:** Data cannot be shared publicly because of they were obtained under a data sharing agreement. Data are available from the Canadian Primary Care Sentinel

## Abstract

Diabetes mellitus type 2 is increasingly being called a modern preventable pandemic, as even with excellent available treatments, the rate of complications of diabetes is rapidly increasing. Predicting diabetes and identifying it in its early stages could make it easier to prevent, allowing enough time to implement therapies before it gets out of control. Leveraging longitudinal electronic medical record (EMR) data with deep learning has great potential for diabetes prediction. This paper examines the predictive competency of deep learning models in contrast to state-of-the-art machine learning models to incorporate the time dimension of risk. The proposed research investigates a variety of deep learning models and features for predicting diabetes. Model performance was appraised and compared in relation to predominant features, risk factors, training data density and visit history. The framework was implemented on the longitudinal EMR records of over 19K patients extracted from the Canadian Primary Care Sentinel Surveillance Network (CPCSSN). Empirical findings demonstrate that deep learning models consistently outperform other state-of-the-art competitors with prediction accuracy of above 91%, without overfitting. Fasting blood sugar, hemoglobin A1c and body mass index are the key predictors of future onset of diabetes. Overweight, middle aged patients and patients with hypertension are more vulnerable to developing diabetes, consistent with what is already known. Model performance improves as training data density or the visit history of a patient increases. This study confirms the ability of the LSTM deep learning model to incorporate the time dimension of risk in its predictive capabilities.

## Author summary

Diabetes is a growing problem around the world and yet it is preventable. A small percentage of people are at higher risk of developing diabetes. Detecting those at highest risk early and offering them early treatment could go a long way to slowing down the growth of diabetes and reverse the trend of severe complications of diabetes. One of the barriers to

Surveillance Network (https://www.cpcssn.ca) for researchers who meet the criteria for access to confidential data.

**Funding:** This research was partially supported by a NSERC Discovery Grant 2019-24 held by author AG. The funders had no role in study design, data collection and analysis, decision to publish, or preparation of the manuscript.

**Competing interests:** The authors have declared that no competing interests exist.

early detection is our inability to take into account the risk that accumulates over time. Someone who has had elevated blood sugar for 5 years has more risk than someone who has only had it for 1 year, yet all current prediction models only take into account the blood sugar and not the time element. This paper reports on our research with artificial intelligence models that can take into account the time element of risk.

## 1. Introduction

Diabetes mellitus type 2 (T2D) is a chronic disease that is growing in prevalence rapidly and is increasingly being called a preventable pandemic [1]. T2D is associated with long term chronic damage and dysfunction of organs particularly the heart, kidneys, eyes and blood vessels [2]. As reported by the International Diabetes Federation, 537 million individuals have diabetes globally, and this number is expected to increase to 783 million by the year 2045 [2]. T2D is the cause of 1.6 million deaths every year and is the seventh major cause of death. Global health care expenditure is currently US $966 billion and expected to increase to $1.054 trillion by 2045 [2]. T2D can be delayed or prevented with appropriate proven interventions. A global meta-analysis of studies showed that diabetes prevention programs had a 3% absolute risk reduction in incidence of diabetes in persons at risk of developing T2D [3]. However, current screening and treatment approaches are inadequate for large scale diabetes prevention because fasting blood sugar (FBS) and hemoglobin A1c (A1c), the screening methods in most wide-spread use, are not sensitive enough, leaving many at risk undetected and are not specific enough and over diagnose the condition [4]. Better methods of detecting diabetes risk early are needed. Since there is currently no cure for diabetes, only early detection and prevention efforts can lessen its long-term complications. Recent availability of data from electronic health records (EMRs) in conjunction with predictive modeling has made it possible to recognize individuals with elevated risk of T2D earlier, more accurately and at greater scale. This has led to the publication of several studies on predicting diabetes using EMR data (Table 1).

A key challenge of current models is the inability to account for accumulated risk that patients experience over time. For example, a patient who has had a blood pressure of 145/90 for 10 years does not have the same risk as a similar individual who has had the identical blood pressure for only 1 year. Yet, most models cannot distinguish between the two and predict the same risk for both patients [12,13].

This study aimed to use deep learning models with memory features to assess the usefulness of artificial intelligence models to take into account the time-dependent nature of cumulative risk. We compare deep learning models with base line machine learning models to forecast

**Table 1. Studies of diabetes prediction in a variety of datasets.**

| Reference | Data set | Methods | Results |
|---|---|---|---|
| Razavian et al [5] | 4.1 million individuals, Pennsylvania | Logistic Regression | AUC of 0.80 |
| Krishnan et al [6] | 330,000 patient dataset, US health insurance | Logistic Regression | AUC of 0.80 |
| Choi et al [7] | 8454 patients, Korean University Guro Hospital | Logistic Regression, K-NN, QDA, LDA | AUC of 0.78 using Logistic Regression |
| Parveen et al [8] | CPCSSN (4403 patient records used) | J48 Decision Tree, Naïve Bayes | AUC of 0.79 using Naïve Bayes |
| Pradhan et al [9] | Pima Indian Diabetes database (768 patients) | ANN, K-NN, Decision Tree, Naïve Bayes | Accuracy of 85% using ANN |
| Sisodia et al [10] | Pima Indian Diabetes database (768 patients) | Decision Tree, SVM, Naïve Bayes | Accuracy of 76.3% using Naïve Bayes |
| Lai et al [11] | CPCSSN (13309 patient records used) | Logistic Regression and Gradient Boosting Machine | AUC 0f 0.84 using Logistic Regression |
| This work | CPCSSN (19181 patient records used) | Deep Learning Models (CNN, LSTM, CNN-LSTM) | Accuracy of 91.6% using CNN-LSTM |

T2D using EMR data extracted from the Canadian Primary Care Sentinel Surveillance Network (CPCSSN). The work also focuses on identifying the most precise deep learning model and critical features for predicting future onset of T2D. Model performance is examined as a function of critical features, various risk factors, training data density and length of visit history.

The rest of this paper is organized as follows: Section 2 describes our methodology and provides an overview of models used in this study. Section 3 presents the results. The discussion and limitations are presented in Section 4 and Section 5 provides our conclusion.

## 2. Methodology

The proposed framework is divided into data collection, data preparation, pre-processing, train-test split, prediction models, quantifying features, and performance evaluation. A schema for diabetes prediction is shown in Fig 1.

The first step was to collect and preprocess the visit records of patients. After preprocessing, records were divided into 80% training records and 20% testing records. Training records were used to train the model to learn the hidden patterns for patients with T2D and normal individuals. Finally, comparative performance analysis was conducted on the remaining 20% of test records using evaluation metrics. A complete description of the phases is presented below.

### 2.1 Dataset and attributes

The data set for this research is collected from the Canadian Primary Care Sentinel Surveillance Network (CPCSSN) from 1998 to 2015. The data set consisted of 368,790 visit records of 19,181 individuals; each visit has 14 features that include non-sequential demographics information (patient id, age, sex,) clinical observations (body mass index (BMI), systolic blood pressure (sBP), lab results (hemoglobin A1c (A1c), fasting blood sugar (FBS), low density lipoprotein (LDL), high density lipoprotein (HDL), total cholesterol (TC), triglycerides (TG) and diagnoses (hypertension (HTN), osteoarthritis (OA), chronic obstructive pulmonary disease (COPD), depression). These features describe the T2D history of the patient.

The study sample is comprised of 7715 diabetic and 11466 non-diabetic patients. Among them 57.5% are female and 42.2% are male. Absolute statics of the dataset is presented in Table 2, Figs 2 and 3

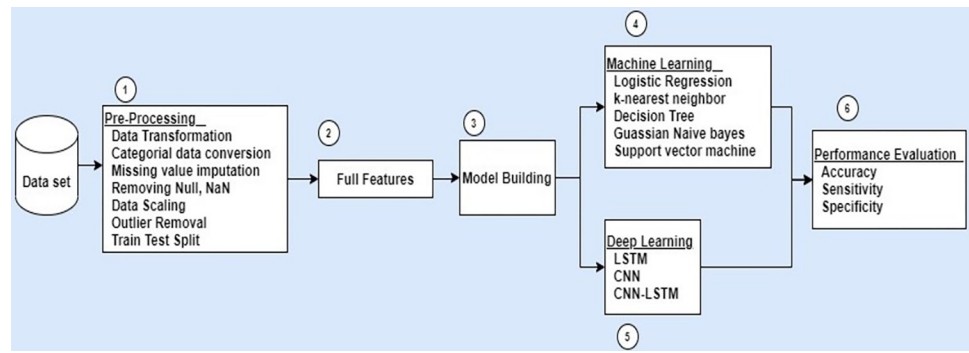

**Fig 1. Analytic framework for T2D prediction study.**

**Table 2. Statics of data.**

| Demographic Information | N (%) |
|---|---|
| Sample | 368790 |
| Patients | 19181 |
| Female patients, samples (%) | 11022 (57.4) |
| Male patients, samples (%) | 8159 (42.5) |
| Age, mean ± SD, Years | 65 ±12.5 |
| **Clinical Observation** | |
| sBP, mean ± SD, mm Hg | 132.05±17.1 |
| BMI, mean ± SD, kg/m2 | 30.71±9.64 |
| **Lab Values** | |
| FBS, mean ± SD, mmol/L | 6.11±1.58 |
| A1C, mean ± SD, mmol/L | 6.4±0.93 |
| TG, mean ± SD, mmol/L | 1.63±1.01 |
| HDL, mean ± SD, mmol/L | 1.35±0.4 |
| LDL, mean ± SD, mmol/L | 2.7±1.05 |
| Cholesterol mean ± SD, mmol/L | 4.79±1.23 |
| BMI, mean ± SD, kg/m2 | 30.71±9.64 |
| **Diagnosis** | |
| Hypertension, N (%) | 13760 (71.7) |
| COPD, N (%) | 213 (1.11) |
| OA, N (%) | 7268 (37.8) |
| Depression, N (%) | 4500 (23.4) |

## 2.2 Data preparation

To prepare the data for diabetes prediction, the visit sequence of diabetic and non-diabetic individuals was prepared separately. The visit history of a diabetic patient from the first visit

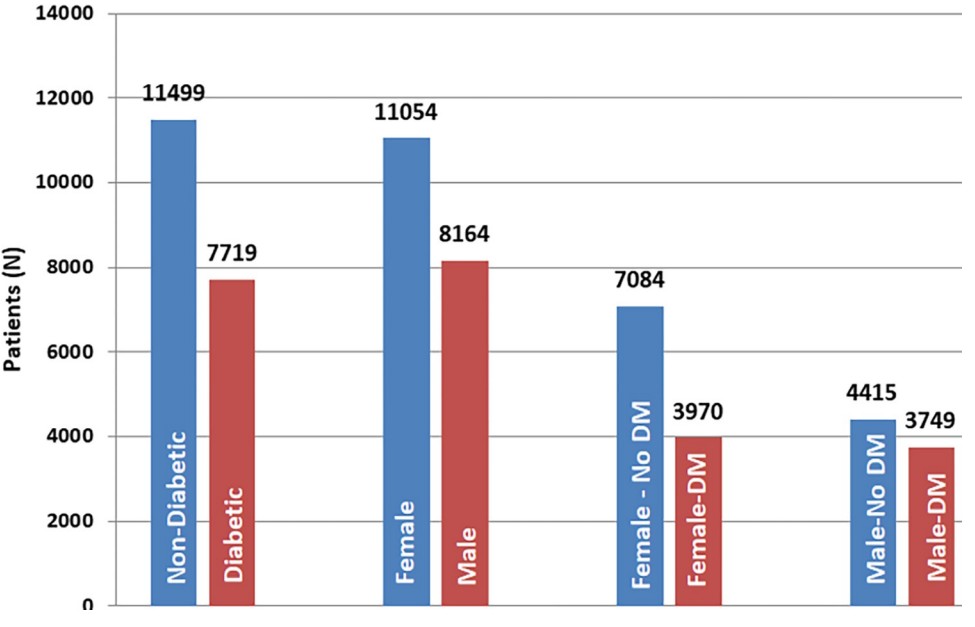

**Fig 2. Description of data.**

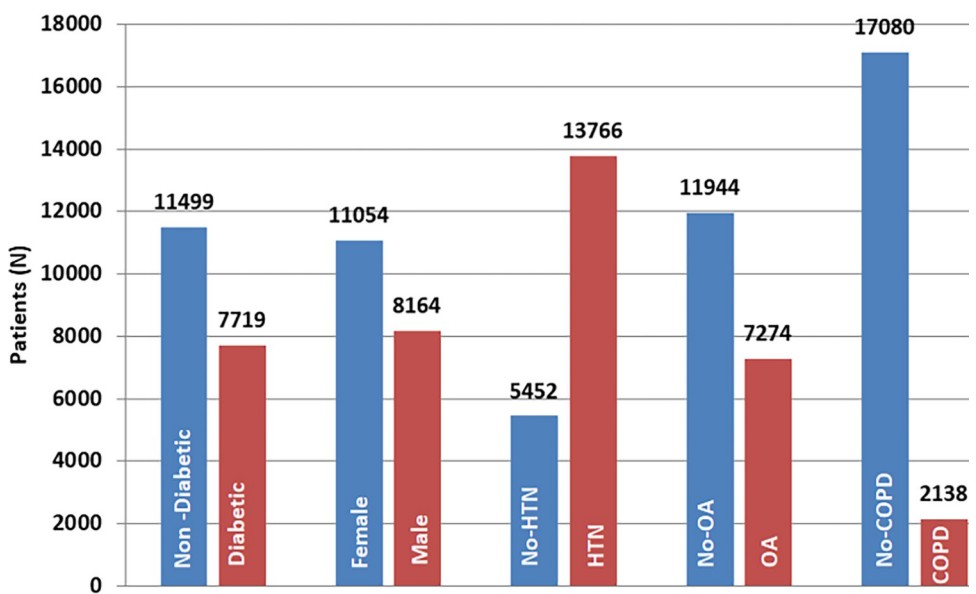

**Fig 3. Statistical distribution of data.**

up to the diabetes incident visit ($N_d$) were retained; visit history after the diabetes incident visit was discarded. However, for non-diabetic individuals, all visit records of the individual from the first to the last visit (N) were retained.

## 2.3 Pre-processing

The intent of preprocessing is to transform the data into a suitable format for training a model. Preprocessing comprises of grouping, sorting, data conversion, missing value imputation, data transformation, outlier removal, data scaling, and feature vector construction.

The visit sequence of every patient is assembled and sorted in ascending order to ensure that the visit history of the patient is sequential. After sorting, categorical variables (e.g., sex) were transmuted into numeric values. Missing values were imputed using mean imputation. Clinical observations or lab results that were outside the standard deviation of (+3 SD and -3 SD) from their mean were considered outliers and were discarded from the visit sequence of a patient. Data scaling was applied to ensure all lab values and clinical observations scaled to the same range between 0 and 1. Feature vectors were constructed and entered into the model to analyze the visit sequence trend for prediabetic individuals. Feature vectors for prediabetic individuals contain the visit records from the first visit to the visit prior to diabetes incidence $(N_d-1)^{th}$. Feature vectors for non-diabetic individuals contains the visit record from the first visit up to the second last visit (N-1).

## 2.4 Train test split

Preprocessed records were divided into 80% training records and 20% testing records. Training records were used to train a model and learn to differentiate between the hidden visit sequence pattern for normal and prediabetic individuals. Test records were utilized to perform comparative performance analysis.

## 2.5 Models

This section provides a brief discussion on various machine learning and deep learning models that were used in this study.

***Logistic Regression (LR):*** A statistical method that implements a logistic function or sigmoid function to model the data for prediction [14] and predicts output as a function of input variables. L2 regularization is used to prevent overfitting of the model.

***Support vector machine (SVM):*** Discovers the hyper plane in an N-dimensional plane (N is number of features), to segregate patients into diabetic and non-diabetic patients and predicts the output [14]. New patients are evaluated according to the decision boundaries, to analyze the portion of the hyper plane on which the patient lies (diabetic or non-diabetic) [14,15]. A linear kernel is used with SVM.

***Decision Tree (DT):*** A flow chart-like structure of a tree constructed from top (root) to bottom (leaf). A decision tree segregates the data into subsets that forms the basis for prediction. DTs predict the output based on decisions on input variables [10,16]. Based on the conditions or internal nodes, a tree is split into various branches or edges. The branch where a tree cannot be split further is the decision or leaf (i.e., a patient will be diabetic or non-diabetic).

***Gaussian Naïve Bayes (GNB):*** Works using probabilistic measures based on Bayes theorem and continuous valued features follow a Gaussian normal distribution. It works with the strong assumptions of independence and equal importance of features (same weights for all features) for prediction [17].

**A. Long short term memory.** Sophisticated gates and feedback loops in LSTM confer a distinctive potential to memorize past values and to remember long term dependencies. [18–20]. LSTM encompasses cell state $c_t$, and three gates: a forget gate $f_t$, an input gate $i_t$, and an output gate $o_t$. The cell state (memory unit) is the predominant part that retains the information for a short interim period. Gates frequently eliminate, keep, upgrade and circulate the information around the cell state [18,19] to evade long term dependency [21]. The forget gate (Eq 1) evaluates which information to eradicate from the cell state. The input gate (Eq 2) determines which new information from the latest input should be included to the cell state. A candidate cell state $\tilde{c}_t$ is initiated while a tanh function (Eq 3) is used to determine how much information from the old cell state $c_{t-1}$ should be preserved. The cell state combines the old cell state $c_{t-1}$ with the forget gate $f_t$ and the new cell state $c_t$ with an input gate $i_t$ (Eq 4) to upgrade the cell state from $\tilde{c}_t$ to $c_t$. Finally, the output gate $o_t$ (Eq 5) elects which information should be output from the cell state. A tanh function is applied to the updated cell state $c_t$ to ascertain which information to eliminate or append. The accumulating tanh function with the output gate induces a new hidden state $h_t$ (Eq 6) to perform the prediction. The Adam optimization algorithm is used as a stochastic optimization algorithm to train the deep learning model. The number of epochs, which is a hyperparameter of the optimization algorithm, is set to 30, which represents the number of passes through the training dataset.

$$f_t = \sigma(W_f x_t + U_f h_{t-1} + b_f) \tag{1}$$

$$i_t = \sigma(W_i x_t + U_i h_{t-1} + b_i) \tag{2}$$

$$\tilde{c}_t = tanh(W_c x_t + U_c h_{t-1} + b_c) \tag{3}$$

$$c_t = f_t \odot c_{t-1} + i_t \odot \tilde{c}_t \tag{4}$$

$$o_t = \sigma(W_o x_t + U_o h_{t-1} + b_o) \tag{5}$$

$$h_t = o_t \odot tanh(c_t) \tag{6}$$

**B. Convolutional neural network.**   CNN can find hidden patterns and temporal relationships within a dataset [20,21]. CNN incorporates a convolution layer, a pooling layer and a fully connected layer. The convolutional layer implements a convolution of input with a kernel or filter to generate a feature map and elapse to the activation layer (Eq 7). The convolutional layer output is fed to the pooling layer (Eq 8) to down sample (reduce) the number of features. The fully connected layer is comprised of a rectified linear function (ReLU) or SoftMax function. The output of the pooling layer becomes the input to the fully connected layer to carry out prediction. Here also the Adam optimization algorithm is used with the number of epochs set to 30.

$$Y_a = \sigma(W_a * X_a + B_a) \tag{7}$$

$$Y_i = max(y_i) \tag{8}$$

**C. Hybrid CNN-LSTM.**   In a hybrid CNN-LSTM, the CNN extracts temporal features while the LSTM memorizes the long-term dependency [22]. Extracted CNN features are input to LSTM, and LSTM analyzes the feature maps to perform the desired prediction.

## 2.6 Feature importance

Feature selection methods are employed to examine the most significant features that contribute to diabetes prediction [23–25]. Our framework employs univariate feature selection, and feature importance scores to obtain optimal subsets of features for diabetes. Univariate feature selection employs a statistical test (chi-square) to calculate chi-squared value and ranks the features according to their importance in predicting diabetes. In the feature importance score, the importance score is allotted using the classification and regression tree (CART) technique to individual features and features are ranked based on their potential to predict diabetes.

## 2.7 Evaluation approach

The discriminatory potential of the proposed models is evaluated using accuracy, sensitivity specificity, precision and F1-score, which are common metrics used to evaluate the performance of AI-based models [26,27]. Accuracy is the proportion of accurately predicted patients (Eq 9). Sensitivity is the proportion of patients that will be diabetic and are accurately predicted as diabetic (Eq 10). Specificity is the proportion of patients that will be non-diabetic and are accurately predicted as non-diabetic (Eq 11). In addition, the metrics of precision (Eq 12) and F1-score (Eq 13) are also provided, where the former measures the quality of the prediction, and the latter measures the accuracy of the model.

$$Accuracy = \frac{TP}{TP + TN + FP + FN} \tag{9}$$

$$Sensitivity = \frac{TP}{TP + FN} \tag{10}$$

$$Specificity = \frac{TN}{TN + FP} \tag{11}$$

$$precision = TP/(TP + FP) \tag{12}$$

$$F_{1\_}score = TP/(TP + 1/2(FP + FN)) \tag{13}$$

## 3. Experimental results

### 3.1 Performance of models

The predictive efficacy of deep learning models in contrast with baseline machine learning models for enhanced diabetes prediction is presented in Table 3 and Fig 4.

Comparison of the accuracy of the baseline machine learning models with deep learning models demonstrates that deep learning models outperform state-of-the-art machine learning with more than 91% prediction accuracy. SVM also performed well with an accuracy of 90%. Linear regression (LR) with an accuracy of 89.6% provides satisfactory performance. Gaussian Naïve Bayes (GNB) and Decision Trees (DT) with accuracy of 84.9% and 85.5%, respectively, show relatively poor performance.

In terms of sensitivity, deep learning models along with LR achieve the highest range of sensitivity with more than 85%. The next best models in terms of sensitivity are SVM and DT with (sensitivity of 84.6% and 82.4%). Whereas GNB, presents the worst sensitivity of 77.6%.

Comparison of the specificity of machine learning and deep learning model shows that the deep learning model together with SVM provides the highest range of specificity greater than 95%. GNB and LR also provide acceptable specificity of 90.9% and 92.6%. Contrarily DT has the lowest specificity performance.

Comparison of precision demonstrates that deep learning model achieves the highest range of precision with more than 91%, with SVM following close behind. LR and GNB have adequate precision, whereas DT represents the worst precision at 80.5%.

Comparison of F1 scores of the models shows that the deep learning models have F1 scores greater than 89% while machine learning models perform relatively poorly.

Overall, the deep learning models LSTM, CNN and CNN-LSTM exhibit the best diabetes prediction in contrast to baseline machine learning models. SVM performance is comparable to the deep learning models.

### 3.2 Feature importance

The relative importance of features for diabetes prediction is presented in Figs 5 and 6. Feature scores are calculated using the chi square test while importance scores are assigned using classification and regression tree (CART) analysis.

It is clearly seen that FBS, A1c and BMI are the predominant features for diabetes prediction as shown in Figs 5 and 6. An increase in FBS or A1c levels maximizes the probability of developing diabetes. The next most important feature of diabetes is BMI, which means that individuals with higher BMI are at greater risk of developing diabetes. The lipid profile (LDL, HDL, TC, TG) are also important risk factors and contribute to diabetes prediction.

**Table 3. Comparison of model performance.**

|  | Accuracy (%) | Sensitivity (%) | Specificity (%) | Precision (%) | F1-Score |
|---|---|---|---|---|---|
| **LR** | 89.6 | 85.1 | 92.6 | 88.6 | 86.8 |
| **DT** | 84.9 | 82.4 | 86.6 | 80.5 | 81.5 |
| **GNB** | 85.5 | 77.6 | 90.9 | 85.1 | 81.2 |
| **SVM** | 90 | 84.6 | 94.4 | 91.0 | 87.7 |
| **LSTM** | 91.5 | 86.6 | 94.9 | 91.9 | 89.1 |
| **CNN** | 91.7 | 86.7 | 95.0 | 92.2 | 89.3 |
| **CNN-LSTM** | 91.6 | 85.7 | 95.6 | 93.0 | 89.2 |

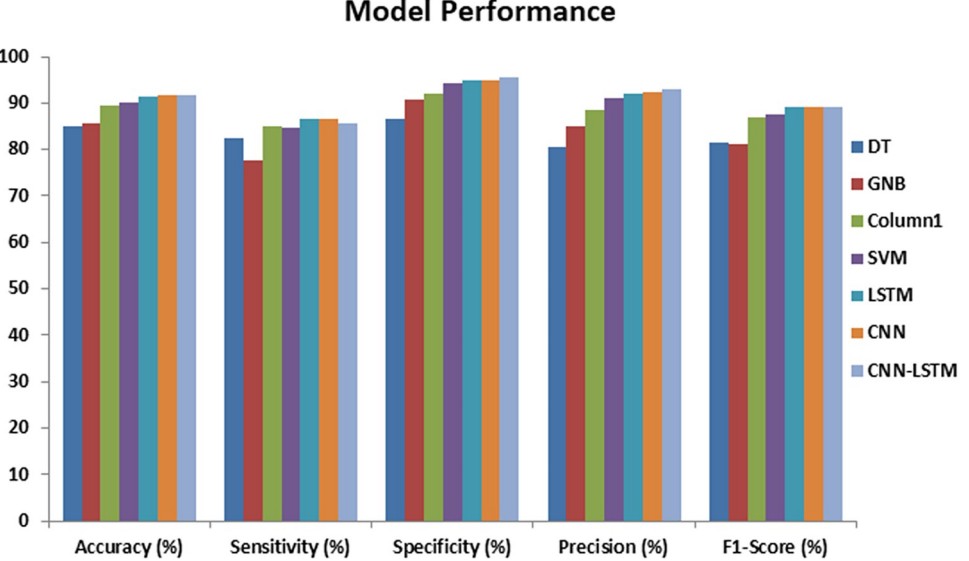

**Fig 4. Model Performance.**

### 3.3 Model performance of selected subset of features

Multiple experiments are implemented to explore the effect of the foremost feature, on data driven deep learning models (LSTM, CNN-LSTM) as summarized in Table 4. Both models exhibit excellent prediction accuracy of more than 86% with only FBS and A1c, evincing the exceptional predictive potential of these features. The next best subset of features, BMI and lipid profile attain an adequate prediction accuracy of 71.1% and 72.5%, respectively.

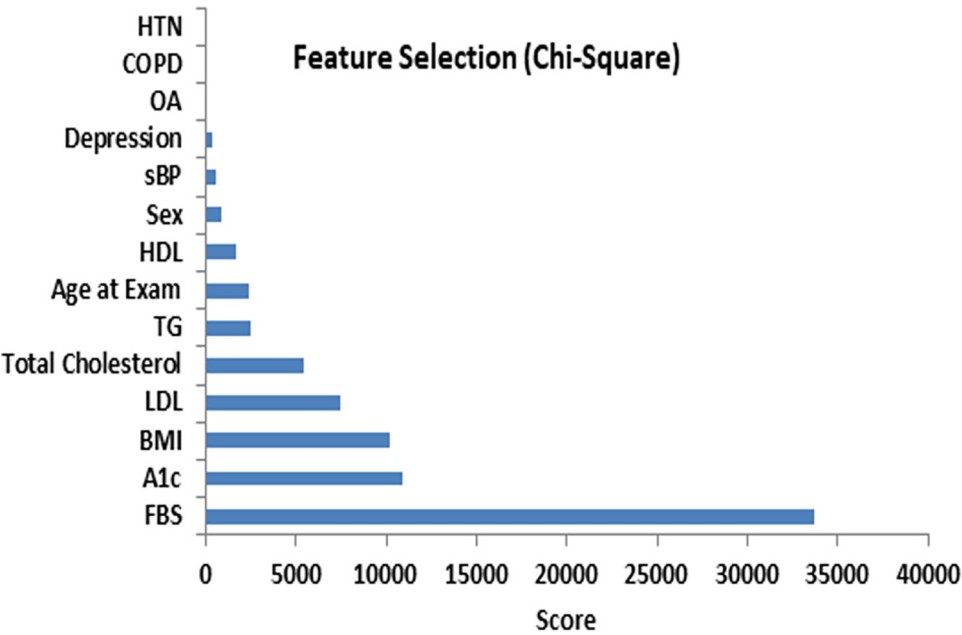

**Fig 5. Feature ranking (chi-square test).**

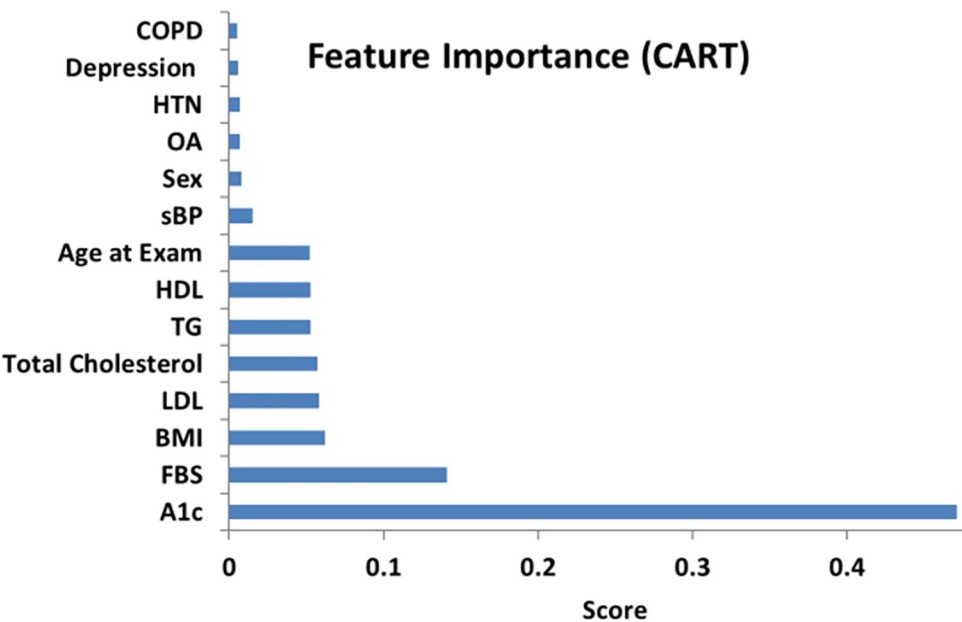

**Fig 6. Feature importance (CART).**

## 3.4 Model performance for critical risk factors

Model performance was examined to analyze how advanced deep learning models (LSTM and CNN-LSTM) perform for various ranges of BMI, age, and sBP and HTN.

Diabetes prevalence for BMI ranges is shown in Table 5. Both the deep learning models (LSTM and CNN-LSTM) present the highest prediction accuracy of 89.8% and 89.7% for obese patients in contrast to normal patients with much lower, but still very respectable, model performance of 87.6% and 87.9%, respectively shown in Fig 7.

Table 6 summarizes the diabetes prevalence for different age groups. Both the deep learning models (LSTM and CNN-LSTM) exhibit the preeminent prediction accuracy of over 90% for middle aged group people (40 to 60). Although model performance declines for elder patients, they are still better than machine learning models as shown in Fig 8.

Diabetes prevalence with hypertension is reported in Table 7. Both the models (LSTM and CNN-LSTM) provide excellent prediction accuracy, beyond 90% for patients with hypertension and prehypertension, as shown in Fig 9.

## 3.5 Model performance with diversified training data density

The influence of training data size on model performance is represented in Table 8. Training size was gradually varied from 484 patients to 8712 patients. Model performance for LSTM

**Table 4. Model performance for subset of features.**

| Features | LSTM (%) | CNN-LSTM (%) |
|---|---|---|
| A1c | 88.2 | 89.7 |
| FBS | 86.4 | 87.8 |
| BMI, LDL, HDL, Cholesterol, TG, Age | 71.1 | 72.5 |
| Sex, sBP, HTN, Depression, OA, COPD | 66.1 | 66.7 |

**Table 5. Comparison of model performance for various BMI ranges.**

| BMI | Test | LSTM (%) | CNN LSTM (%) |
|---|---|---|---|
| > = 18 & < = 25 Normal | 463 | 87.6 | 87.9 |
| > = 25 & < = 30 Overweight | 887 | 89.1 | 89.5 |
| > 30 Obese | 953 | 89.8 | 89.7 |

and CNN-LSTM, improved significantly from 86% to 91% with an increase in training data size, as shown in Fig 10.

### 3.6 Model performance as a function of visit history

Model performance in relation with heterogeneity of visit history of patients varied from first visit to ninth visit and is delineated in Table 9 and Fig 11. Significant enhancement in

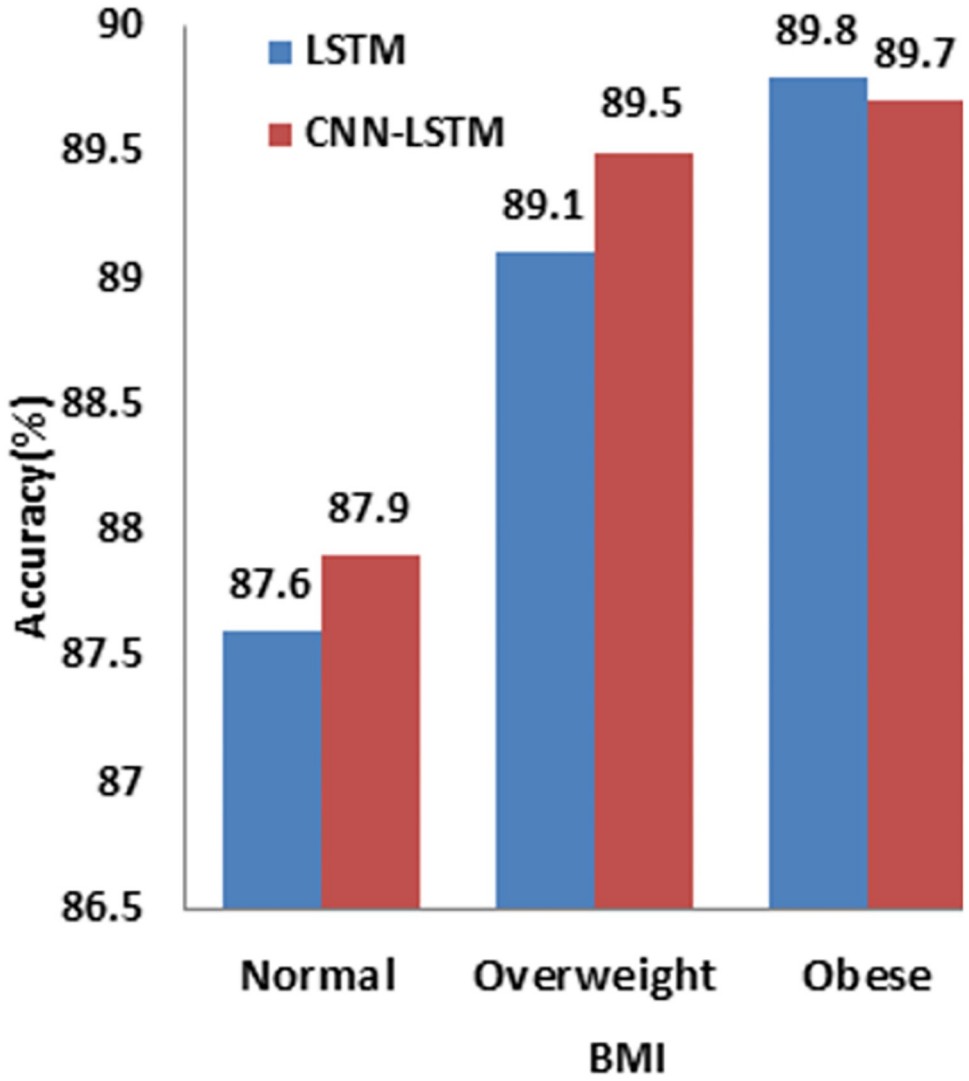

**Fig 7. Comparison of model performance for various BMI ranges.**

Table 6. Comparison of model performance for different age group patients.

| Age | Test | LSTM (%) | CNN LSTM (%) |
|---|---|---|---|
| **40–50** | 261 | 89.65 | 90.4 |
| **50–60** | 591 | 90.1 | 90.1 |
| **60–70** | 805 | 87.8 | 88 |
| **70–80** | 689 | 85.3 | 86.7 |
| **80–90** | 275 | 86.5 | 88 |

prediction accuracy for both the models is observed from 87% to 91% by utilizing longer visit history of patients (up to 9 visits).

## 4. Discussion

Early prediction of diabetes onset is important for all health care systems, as diabetes is now considered a modern preventable pandemic. Leveraging longitudinal EMR data with deep learning can detect individuals at high risk of developing diabetes for early intervention that could delay or even prevent the onset of diabetes. State of the art machine learning algorithms which are reported on extensively in the literature for predictive analysis cannot capture long term sequences and temporal relations.

It is worth noting that, of all the examined state of the art machine and deep learning models, deep learning models (LSTM, CNN and CNN-LSTM) outperform the baseline machine learning models (Table 3) due to their distinctive potential to extract temporal relations. In contrast to widely used machine learning models, LS+TM has greater potential to extract complex information from time series data due to their hierarchical structure. Moreover, the feedback connection in LSTM helps to capture the sequential information in data and to forecast

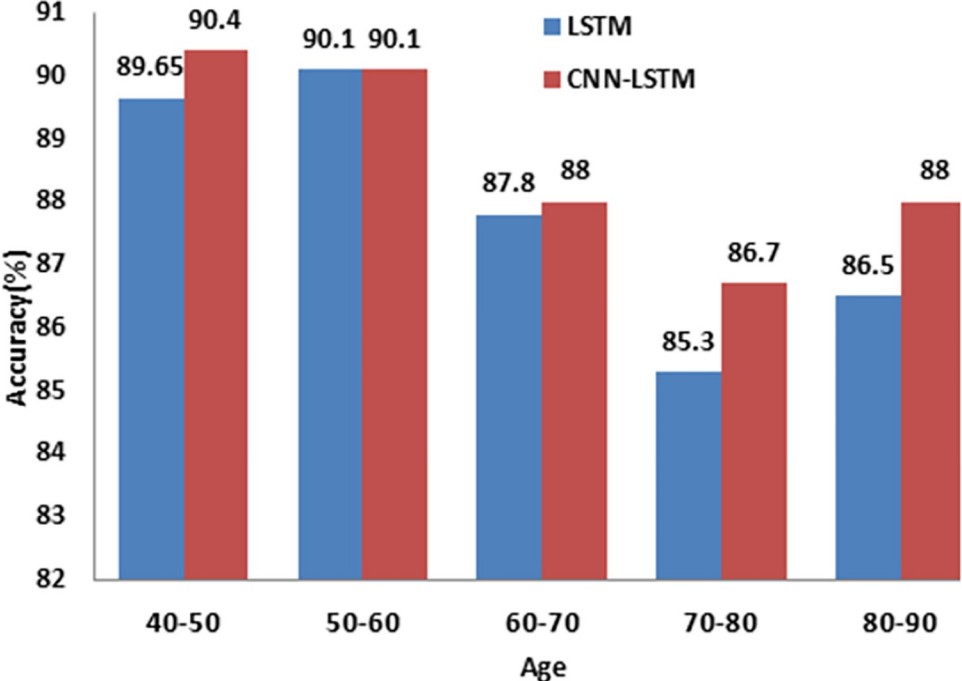

Fig 8. Comparison of model performance for different age group patients.

Table 7. Comparison of model performance for sBP and normal patient.

|  | Test | LSTM (%) | CNN LSTM(%) |
| --- | --- | --- | --- |
| <120 Normal | 626 | 88.0 | 88.6 |
| 120–129 Pre-hypertension | 1502 | 90.2 | 90.4 |
| 130–139 HTN 1 | 455 | 90.1 | 90.5 |

the future based on past data. The gating structure of LSTM controls the flow of information into the cell and provides a memory for long term dependencies in time series data. It is thus that deep learning models like LSTM can better utilize temporal features of EMR data than traditional machine learning models and could be used to enhance other clinical predictive tasks. Furthermore, of all the base line machine learning models, SVM also has considerable predictive competency.

There are two main benefits of using deep learning models for the prediction of diabetes. The first is the ability to take into account the temporal nature of risk, which accumulates over time to predict diabetes with a higher accuracy. The second is the ability of the model to work when limited data may be available. The proposed method shows a less than 5% decrease in accuracy when the size of the training data is decreased from 90% to 5%. This has implications for predicting diabetes with higher accuracy in situations when data is limited. Limitations of

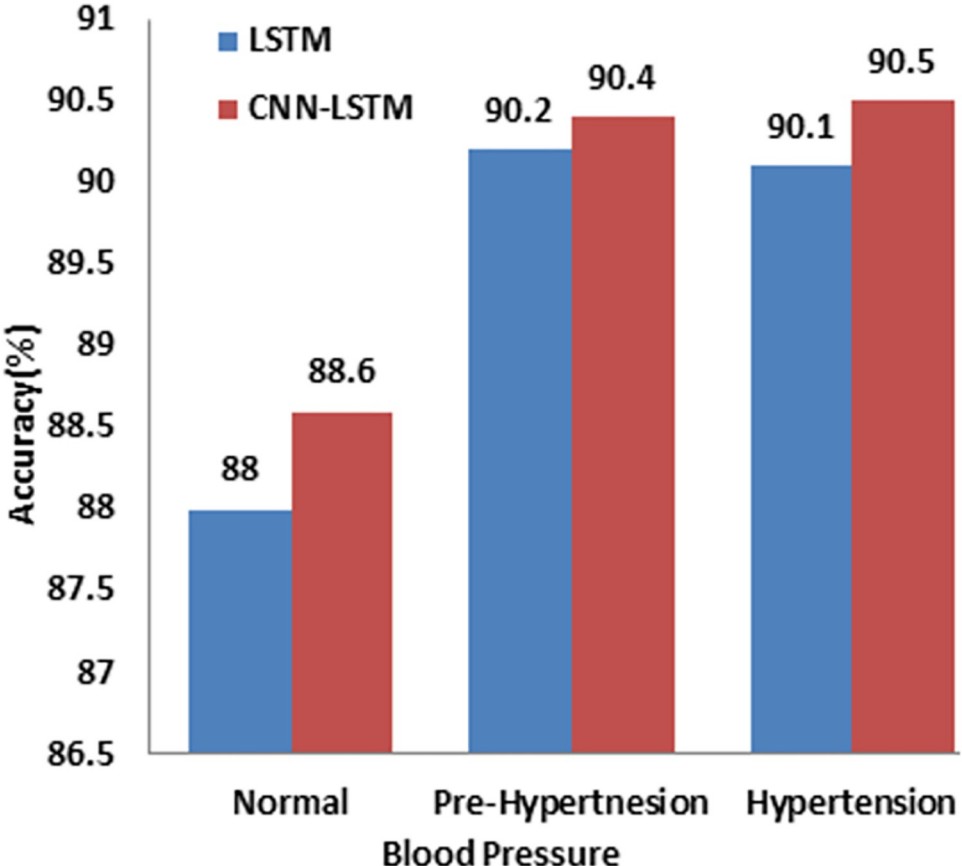

Fig 9. Comparison of model performance for normo-tensive, prehypertensive and hypertensive individuals.

**Table 8. Comparison of model performance for different training size.**

| Train | Test | LSTM (%) | CNN-LSTM (%) |
|---|---|---|---|
| 8712 (90%) | 969 (10%) | 91.2 | 91.1 |
| 7744 (80%) | 1937 (20%) | 91.1 | 91.4 |
| 6776 (70%) | 2905 (30%) | 90.3 | 90.5 |
| 5808 (60%) | 3873 (40%) | 89.7 | 90.1 |
| 4840 (50%) | 4841 (50%) | 89.4 | 89.5 |
| 3872 (40%) | 5809 (60%) | 89.1 | 89.3 |
| 2904 (30%) | 6777 (70%) | 88.6 | 88.8 |
| 1936 (20%) | 7745 (80%) | 87.9 | 88.5 |
| 968 (10%) | 8713 (90%) | 87.6 | 87.8 |
| 484 (5%) | 9197 (95%) | 86.8 | 86.6 |

the study include lack of socio-economic data, family history, dietary habits, physical activity, sleep patterns, psychosocial stress levels and microbiome data, which are known factors in the development of obesity and diabetes.

## 5 Conclusion

This study compares the predictive strength of deep learning models with machine learning models. The intent is to identify the most precise deep learning model that provides temporal features and the most significant features for diabetes prediction. Model performance was assessed for critical features, risk factors, training data density, and visit history of a patient. The results exhibit that deep learning models offer superior diabetes prediction with enhanced performance accuracy above 91%. The predictive competency analysis of features exhibits significant predictive potential for key features such as FBS, A1c and BMI. Risk factor analysis indicates that obese, middle aged and hypertensive individuals are more susceptible to diabetes, in keeping with known medical knowledge, but not used quantitatively in current clinical practice to predict future onset of diabetes. The magnitude of training data and length of visit history of a patient substantially improves model performance. Prediction accuracy increases

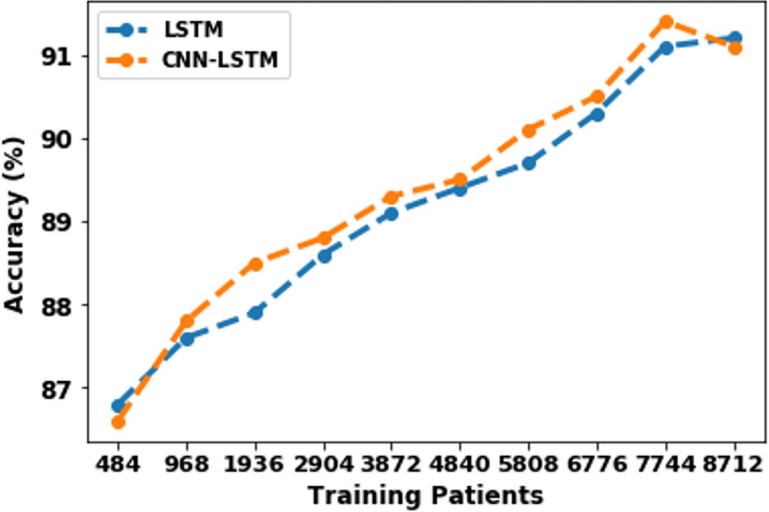

**Fig 10. Comparison of model performance for different training data size.**

**Table 9. Comparison of model performance for different training size.**

| Number of Visits | LSTM | CNN-LSTM |
|---|---|---|
| 1 | 87.5 | 87.2 |
| 2 | 87.8 | 87.8 |
| 3 | 88.4 | 88.1 |
| 4 | 88.7 | 88.7 |
| 5 | 89.0 | 89.0 |
| 6 | 89.3 | 89.3 |
| 7 | 90.2 | 90.5 |
| 8 | 90.8 | 90.8 |
| 9 | 91.9 | 91.3 |

as training data density increases or the number of visits increases. Excellent prediction accuracy is attained with maximal training data density (8712 patients) and substantial visit sequence (9 visits for patient).

This study makes the following contributions to current knowledge: 1) confirms that the LSTM deep learning model incorporates the time component of risk into its predictions, which has been difficult to achieve to date with other models. 2) Incorporates known qualitative variables, such as obesity, age and co-morbidities into its predictive capabilities, thereby increasing the sensitive and specificity of diabetes prediction significantly.

An interesting direction that future work may take is the development of a framework for a recommendation system for patients who are at high risk of developing diabetes. The important risk factors for diabetes could be further investigated in the context of diabetes prediction. The models presented in this paper could be adapted to other diseases and datasets.

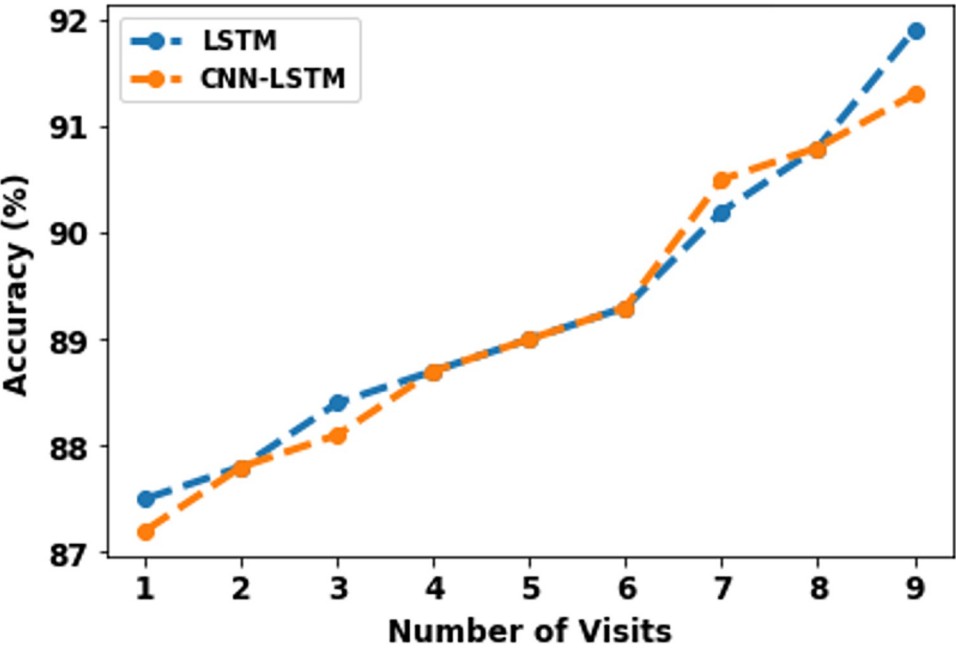

**Fig 11. Comparison of model performance for different number of visits.**

## Author Contributions

**Conceptualization:** Iqra Naveed, Muhammad Farhat Kaleem, Karim Keshavjee, Aziz Guergachi.

**Data curation:** Iqra Naveed, Karim Keshavjee.

**Formal analysis:** Iqra Naveed.

**Investigation:** Iqra Naveed, Muhammad Farhat Kaleem, Aziz Guergachi.

**Methodology:** Iqra Naveed, Muhammad Farhat Kaleem.

**Project administration:** Muhammad Farhat Kaleem.

**Supervision:** Karim Keshavjee.

**Validation:** Karim Keshavjee.

**Visualization:** Iqra Naveed.

**Writing – original draft:** Iqra Naveed.

**Writing – review & editing:** Muhammad Farhat Kaleem, Karim Keshavjee, Aziz Guergachi.

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
