## [Decision Letter · Decision Letter 0]

1 Jun 2023

PDIG-D-23-00121

Artificial intelligence with temporal features outperforms machine learning in diabetes prediction

PLOS Digital Health

Dear Dr. Keshavjee,

Thank you for submitting your manuscript to PLOS Digital Health. After careful consideration, we feel that it has merit but does not fully meet PLOS Digital Health's publication criteria as it currently stands. Therefore, we invite you to submit a revised version of the manuscript that addresses the points raised during the review process.

Please submit your revised manuscript within 30 days Jul 01 2023 11:59PM. If you will need more time than this to complete your revisions, please reply to this message or contact the journal office at digitalhealth@plos.org. Please include the following items when submitting your revised manuscript:

We look forward to receiving your revised manuscript.

Kind regards,

Danilo Pani, Ph.D.

Academic Editor

PLOS Digital Health

Journal Requirements:

Additional Editor Comments (if provided):

According to the Reviewers' comments and my personal judgement, I think the manuscript deserves some more clarifications. Please carefully consider the main comments from them

Reviewers' comments:

Reviewer's Responses to Questions

**Comments to the Author**

1. Does this manuscript meet PLOS Digital Health’s publication criteria? Is the manuscript technically sound, and do the data support the conclusions? The manuscript must describe methodologically and ethically rigorous research with conclusions that are appropriately drawn based on the data presented.

Reviewer #1: Yes

Reviewer #2: Yes

2. Has the statistical analysis been performed appropriately and rigorously?

Reviewer #1: Yes

Reviewer #2: No

3. Have the authors made all data underlying the findings in their manuscript fully available (please refer to the Data Availability Statement at the start of the manuscript PDF file)?

Reviewer #1: Yes

Reviewer #2: Yes

4. Is the manuscript presented in an intelligible fashion and written in standard English?

Reviewer #1: Yes

Reviewer #2: No

5. Review Comments to the Author

Reviewer #1: This paper provides a novel contribution to the field by using deep learning method to perform diabetes prediction.

The proposed method utilizes the temporal relationships among the repeated measurements of the patients information. It outperforms baseline machine learning methods which do not take into account of the temporal information of the data.

The paper presented numeric studies to demonstrate the advantage of the proposed method.

Reviewer #2: I would like to see the following modifications in the revised version that will improve the current work.

1. Summarize the recent works in the form of a table.

2. The model’s performance is evaluated using accuracy, specificity and sensitivity, precision and f1-score but the formulas for precision and F1-score are not defined in section 2.7. include them accordingly.

3. Compare and explain the significance of using machine learning and deep learning in diabetes prediction.

4. Give a comparison of earlier reported work with current work in the Discussion section.

5. What is the benefit of the designed method? Explain

6. The Tuning number of layers in machine and deep learning models is not clear, kindly revise it and add more explanations.

7. Make a statistical test to compare the results against the methods in literature for the data sets presented.

8. Improve the English language for enhanced the quality of the paper.

9. Aggregate articles from the journal " PLOS Digital Health".

6. PLOS authors have the option to publish the peer review history of their article (what does this mean?). If published, this will include your full peer review and any attached files.

**Do you want your identity to be public for this peer review?** For information about this choice, including consent withdrawal, please see our Privacy Policy.

Reviewer #1: No

Reviewer #2: Yes: Dr. Deepti Sisodia

---

## [Decision Letter · Decision Letter 1]

19 Aug 2023

Artificial intelligence with temporal features outperforms machine learning in diabetes prediction

PDIG-D-23-00121R1

Dear Dr. Keshavjee,

We are pleased to inform you that your manuscript 'Artificial intelligence with temporal features outperforms machine learning in diabetes prediction' has been provisionally accepted for publication in PLOS Digital Health.

Best regards,

Danilo Pani, Ph.D.

Academic Editor

PLOS Digital Health

As correctly pinpointed by the Reviewer, the assessment metrics proposed are not statistical tests. I assume the Reviewer was referring to test ON those metrics. Nevertheless, I think the corrections are fair and the manuscript can now be accepted.

Reviewer Comments (if any, and for reference):

Reviewer's Responses to Questions

**Comments to the Author**

1. If the authors have adequately addressed your comments raised in a previous round of review and you feel that this manuscript is now acceptable for publication, you may indicate that here to bypass the “Comments to the Author” section, enter your conflict of interest statement in the “Confidential to Editor” section, and submit your "Accept" recommendation.

Reviewer #2: All comments have been addressed

2. Does this manuscript meet PLOS Digital Health’s publication criteria? Is the manuscript technically sound, and do the data support the conclusions? The manuscript must describe methodologically and ethically rigorous research with conclusions that are appropriately drawn based on the data presented.

Reviewer #2: Yes

3. Has the statistical analysis been performed appropriately and rigorously?

Reviewer #2: No

4. Have the authors made all data underlying the findings in their manuscript fully available (please refer to the Data Availability Statement at the start of the manuscript PDF file)?

Reviewer #2: Yes

5. Is the manuscript presented in an intelligible fashion and written in standard English?

Reviewer #2: Yes

6. Review Comments to the Author

Reviewer #2: REVIEWS (MINOR REVISION)

PDIG-D-23-00121R1 for PLOS Digital Health entitled Artificial intelligence with temporal features outperforms machine learning in diabetes prediction .

Following suggestion need to be addressed:

sensitivity, specificity, precision and F1 scores are not statistical tests but are evaluation measures which are used to evaluate the performance or quality of the model, and these metrics are known as performance metrics or evaluation metrics.

Statistical tests are a way of mathematically determining whether two sets of data are significantly different from each other. There are various statistical tests that can be used, depending on the type of data being analyzed. However, some of the most common statistical tests are chi-squared tests, ANOVA test. t-test, Z-test, chi-square test, binomial test and so on….

7. PLOS authors have the option to publish the peer review history of their article (what does this mean?). If published, this will include your full peer review and any attached files.

**Do you want your identity to be public for this peer review?** For information about this choice, including consent withdrawal, please see our Privacy Policy.

Reviewer #2: **Yes: **Dr. Deepti Sisodia
